Subject Category:
Biology (whole organism)

Subject Areas:
ecology

Keywords:
bumblebees, *Bombus impatiens*, *Crithidia bombi*, goldenrod, pollinator decline, sunflower

Author for correspondence:
George M. LoCascio
e-mail: glocascio@umass.edu

# Pollen from multiple sunflower cultivars and species reduces a common bumblebee gut pathogen

George M. LoCascio[1], Luis Aguirre[2], Rebecca E. Irwin[3] and Lynn S. Adler[2]

[1]Department of Environmental Conservation, and [2]Department of Biology, University of Massachusetts, Amherst, MA 01003, USA
[3]Department of Applied Ecology, North Carolina State University, Raleigh, NC 27695, USA

 GML, 0000-0003-1248-9221; REI, 0000-0002-1394-4946

Pathogens are one of the factors driving pollinator declines. Diet can play an important role in mediating pollinator health and resistance to pathogens. Sunflower pollen (*Helianthus annuus*) dramatically reduced a gut pathogen (*Crithidia bombi*) of *Bombus impatiens* previously, but the breadth of this effect was unknown. We tested whether pollen from nine *H. annuus* cultivars, four wild *H. annuus* populations, *H. petiolarus*, *H. argophyllus* and two *Solidago* spp., reduced *Crithidia* in *B. impatiens* compared to mixed wildflower pollen and buckwheat pollen (*Fagopyrum esculentum*) as controls. We also compared hand- and honeybee-collected pollen (which contains nectar) to assess whether diet effects on pathogens were due to pollen or nectar. All *Helianthus* and *Solidago* pollen reduced *Crithidia* by 20–40-fold compared to buckwheat pollen, and all but three taxa reduced *Crithidia* compared to wildflower pollen. We found no consistent differences between hand- and bee-collected pollen, suggesting that pollen alone can reduce *Crithidia* infection. Our results indicate an important role of pollen diet for bee health and potentially broad options within the Asteraceae for pollinator plantings to manage bee disease.

## 1. Background

Pollination services are critical in ecological and agricultural systems. In the United States, up to 90 crops are pollinated by bees [1] and worldwide, pollinators pollinate about one-third of food crops [2]. Pollinators also fill important ecological niches by aiding wild plant reproduction, contributing to the maintenance of a diverse landscape [3,4]. Since the turn of the twenty-first century, several pollinator taxa have declined, including some bee

species [5–7]. With mounting concerns about these declines [8,9], research on pollinator diseases and their potential mitigation has become a pressing need [10,11].

Most bees rely solely on nectar and pollen as food sources, obtaining lipids and proteins from pollen and sugars from nectar [12]. Wildflower gardens and pollinator strips along agricultural lands are receiving increased attention as mechanisms to provide foraging habitat and nesting sites for pollinators [13]. Flowers can provide not only nutritional benefits but also play a role in mediating bee disease dynamics. Some floral rewards have properties that can reduce parasites [14], which we refer to hereafter as a 'medicinal' trait. If floral rewards of certain plant species are medicinal, this suggests potential benefits if these species are planted in wildflower gardens or pollinator strips. Thus, identifying plants with floral rewards that suppress pathogens could provide non-chemical options to improve pollinator health by incorporating target plant species into agroecosystems and natural habitats.

Studies of sunflower floral rewards (*Helianthus annuus* L.; Asteraceae) indicate that they may play a significant role in pathogen suppression. When compared to other monofloral pollen diets and a wildflower pollen mix, two cultivars of sunflower pollen (*Helianthus annuus* L.; Asteraceae) dramatically suppressed the trypanasomatid intestinal pathogen *Crithidia bombi* in the common eastern bumblebee *Bombus impatiens* and had less dramatic but still significant effects reducing the microsporidian pathogen *Nosema ceranae* in honeybees, *Apis mellifera* [15]. This discovery is consistent with two other studies suggesting that floral rewards from sunflower and related taxa have medicinal properties for bees. For example, ingestion of sunflower honey, which is made of primarily nectar with some pollen, reduced *N. ceranae* and increased survival in honeybees [16]. Additionally, some solitary bees are specialists on Asteraceae pollen [17] and it has been suggested that in *Osmia*, this may be due to pollen reducing brood parasitism [18], although other explanations may also explain these patterns [19]. These discoveries suggest that sunflower and possibly broader Asteraceae pollen have medicinal effects that could help bees resist pathogens or parasites, but the extent of this effect across plant taxa is unknown.

The goal of our study was to assess whether pollen from multiple cultivars and wild populations of *H. annuus*, its congeners and Asteraceae relatives significantly reduced *C. bombi* in *B. impatiens*. Additionally, we compared the effects of hand-collected (i.e. granular) and honeybee-collected pollen because honeybee-collected pollen contains nectar [20,21] and salivary enzymes [22,23] while hand-collected pollen does not. Thus, comparing hand- and honeybee-collected pollen allowed us to ascertain whether medicinal properties are due to pollen, nectar or both. Investigating medicinal effects from a wide range of plant species may provide options for pollinator diets.

# 2. Material and methods

## 2.1. Study system

The common eastern bumblebee, *Bombus impatiens* (Cresson), is a eusocial generalist pollinator with an annual colony life cycle [24], with colonies producing up to 400 workers [25]. *Bombus impatiens* is commonly found in eastern North America, from Maine to Ontario to the eastern Rocky Mountains and south through Florida [26]. Colonies of *B. impatiens* are commercially available.

The intestinal parasite *Crithidia bombi* (Kinetoplastea, Trypanosomatida) is found in wild and commercial *B. impatiens* populations and in other *Bombus* species worldwide [27,28]. *Crithidia bombi* can have varying ranges of parasitism [29], with 49% of bumblebee workers infected in wild colonies in the UK (Goulson *et al.* [30]) and up to 80% in western Massachusetts, USA populations [31]. *Crithidia bombi* is transmitted horizontally during floral visitation [32], and in the hive from one generation of workers to the next via contact with infected faeces [33]. *Crithidia bombi* can reduce *Bombus terrestris* early colony growth rate and successful emergence of hibernating queens [10,11], reduce the production of new daughter queens [30] and interact with starvation to increase mortality by 50% [34]. Furthermore, *C. bombi* reduced *Bombus impatiens*' motor learning rates of flower handling, and foraging rates [35], potentially reducing pollination and foraging efficiency.

Sunflower (*Helianthus annuus*) is a common early successional, self-compatible annual forb native to central North America [36] that is grown commercially for its oilseed and as a cover crop [37], with approximately 22 million ha of *H. annuus* grown for cultivation globally [38]. Moreover, sunflower is also planted on smaller farms in eastern North America for cut flowers and as a novelty or cover crop. Sunflower has relatively low protein compared to other pollen species [39,40] but is actively foraged on by a wide range of bee species [41] including *B. impatiens* [15].

## 2.2. Plant sources and cultivation

We used pollen from nine *H. annuus* cultivars, four populations of wild *H. annuus*, two *Helianthus* congeners, two *Solidago* species and two controls (buckwheat, *Fagopyrum esculentum* and a honeybee-collected wildflower mixed pollen). Hereafter, all 19 pollen treatments are referred to as 'taxa' for simplicity. Pollen from most taxa was collected from plants grown from seed obtained from the USDA Agricultural Research Service through the North Central Regional Plant Introduction Station, which is part of the US National Plant Germplasm System programme. The seeds were sown at the College of Natural Science's greenhouses at the University of Massachusetts-Amherst (electronic supplementary material, table S1) and were grown at the Crop and Animal Research and Education Center in South Deerfield, MA (42°28′45.53″ N 72°34′46.06″ W). We also collected pollen from three taxa outside our field site: *H. annuus* 'Cobalt II' cultivar and *H. annuus* 'Black Oil Seed' cultivar from farms in MA and wild-growing *Solidago* spp. from one population in MA (electronic supplementary material, table S1). For the taxa we did not grow, we used DNA barcoding following established protocols [42] to confirm species identity. Both yellow and orange-coloured *Solidago* pollen had 96% and 97% matches with *Solidago rugosa* and *Solidago canadensis*. Because of the close matches of both colours with both species, we will refer to these taxa as '*Solidago* yellow' and '*Solidago* orange' and both taxa will be considered as potentially both species. *Helianthus annuus* 'Cobalt II' and *H. annuus* 'Black Oil Seed' both produced yellow and orange pollen that we tested separately and were all 96–100% matches with *H. annuus*. We refer to them by their cultivar name and pollen colour. Buckwheat and one source of sunflower pollen used in our original research [15] were obtained from Changge Hauding Wax Industry, China, and the wildflower mix pollen was obtained from Koppert Biological Systems (Linden Apiaries, Howell, MN, USA). We used buckwheat as our single-species comparison to sunflower taxa because buckwheat has a similar protein content as sunflower pollen [39] but results in much higher *C. bombi* infection [15].

## 2.3. Pollen preparation

Pollen collection took place by hand only for five taxa, by honeybees only for eight taxa, and by both methods for three taxa (electronic supplementary material, table S1 and figure S1). Before starting the diet trials, hand- and honeybee-collected pollen was mixed with a 30% 1:1 glucose:fructose sugar solution, reflecting the concentration and sugar ratios in *H. annuus* nectar [43,44]. The ratio of sugar solution to pollen was different between hand- and honeybee-collected pollen to create a dough-like consistency similar across all taxa because hand-collected pollen contained no nectar and thus needed more liquid to reach the same consistency. For hand-collected pollen, we added 43–47% sugar solution by weight, compared to 7–24% sugar solution added to the honeybee-collected pollen. Honeybee-collected pollen can contain up to 40% more sugars by weight than hand-collected pollen [20,45], which roughly corresponds to the 20–40% more sugar solution added to hand-collected compared to honeybee-collected pollen in our experiment.

## 2.4. Inoculum preparation

*Crithidia bombi* were maintained in commercial *B. impatiens* 'source' colonies (Biobest Canada, Leamington, Ontario, Canada) infected with *C. bombi* from wild *B. impatiens* workers collected at Stone Soup Farm in Hadley, MA (42°21′51.93″ N, 72°33′55.88″ W). Every day that we inoculated bees, we prepared fresh *C. bombi* inoculum from 5 to 10 source colony workers using an established protocol [46]. Briefly, the inoculum was prepared by grinding mid- and hindguts in 1.5 ml Eppendorf tubes with 300 µl of one-fourth strength Ringer's solution (Fluka 96724, Sigma-Aldrich, St Louis, MO, USA). The solution was vortexed for 5 s and allowed to settle for 4–5 h at room temperature. After the solution settled, 10 µl samples of the supernatant were placed on a haemocytometer to count *Crithidia* cells. We then used 150 µl samples from 1 to 3 bees to make a mixture diluted with Ringer's solution to achieve 1200 *C. bombi* cells µl$^{-1}$. This solution was mixed with an equal amount of 50% sucrose solution to prepare an inoculum with 600 *C. bombi* cells µl$^{-1}$ in 25% sucrose, which falls within the natural *C. bombi* concentration range in infected faeces [47].

## 2.5. Laboratory trials

During the spring and summer of 2017, workers were isolated from commercially reared laboratory colonies that were confirmed to be free of *C. bombi* via biweekly subsamples of five bees. In total,

17 colonies were used, and each pollen taxon was assessed using at least three colonies. Before inoculation, worker bees were isolated in small vials and starved for 2–3 h. Bees were inoculated individually with 10 µl of fresh *C. bombi* inoculum. Bees were randomly assigned to one of the 19 different pollen treatments and housed individually in plastic 500 ml deli cups with approximately 50 mg (range 40–70 mg) of their treatment pollen and 10 ml of 30% sugar solution, made available by a cotton wick through a hole cut into the top of a 95 mm Petri dish (electronic supplementary material, figure S1C,D). Experimental bees were stored in the dark at 27°C in an incubator. Pollen and sugar solutions were replaced every other day. *Crithidia bombi* reaches a representative level 7 days post-inoculation [47]. Thus, after 7 days, bees were dissected and *C. bombi* was counted as described in '*Inoculum preparation*' above. Radial cell length from the right forewing was measured as a proxy for bee size [48] because previous work suggests bee size affects *C. bombi* cell counts [49]. We ultimately included a total of 650 worker bees (253 bees died, 37 escaped and 13 had damaged wings; see electronic supplementary material, table S1, for treatment sample sizes).

## 2.6. Statistical analysis

All statistical analyses and graphing were conducted using R version 3.3.1 [50]. To examine the effects of pollen treatment on *C. bombi* raw cell counts (cells per 0.02 µl), we used generalized mixed linear models. Owing to the nature of our zero bounded data, we first tested the residuals with a Poisson distribution and checked for over-dispersion. Finding that the data were over-dispersed, we analysed data with a negative binomial error distribution with a log link function using the package *lme4* [51], and calculated least-squares means and standard errors with the package *lsmeans* [52]. We included pollen treatment as a fixed effect, bee size (estimated by radial cell length) as a covariate and date of inoculation and colony of origin as random effects. Upon finding a significant overall effect of pollen treatment, we compared differences among pollen treatments using a Tukey's HSD *post hoc* test. In a separate analysis, we asked whether pollen species differed in their ability to reduce *C. bombi* by pooling pollen treatments into their respective species, or genera in the case of *Solidago* (*H. annuus*, *H. petiolarus*, *H. argophyllus*, *Solidago* spp.). Pollen species instead of taxon was used as a fixed predictor, bee size as a covariate and inoculation date and colony of origin as random effects. We also asked whether hand-collected versus honeybee-collected pollen differed in the ability to reduce *C. bombi* cell counts using a similar analysis, but with collection method as the predictor instead of species, including all taxa in one analysis and a separate analysis only for the three taxa which were collected using both methods. Finally, we used a survival analysis with the package *survminer* [53] to examine whether pollen treatment affected *B. impatiens* mortality rates by comparing our model with and without pollen treatment as the predictor; the model also included bee size, date of inoculation and colony of origin. We removed 50 bees that escaped or had wing damage from our survival analysis. Figures were made with *ggplot2* [54] and *cowplot* [55].

## 3. Results

*Crithidia bombi* cell counts in *B. impatiens* were at least 90% lower in all *Helianthus* and *Solidago* pollen treatments compared to buckwheat pollen (figure 1). All but three taxa (*H. annuus* 'Germany' hand-collected, *H. annuus*, 'wild California' honeybee-collected and *H. petiolaris*) had at least 80% lower *C. bombi* cell counts than the wildflower pollen mix; these differences were significant. Some Asteraceae taxa, such as *Solidago* (yellow) and *H. annuus* 'Germany' (honeybee-collected) had significantly lower cell counts than others, such as *H. annuus* 'wild California' hand-collected and *H. annuus* 'Cobalt II' orange (figure 1). There was a negative relationship between bee size and *C. bombi* counts ($\chi^2_1 = 16.49$, $p < 0.001$), such that larger bees had lower counts across all pollen taxa. In the survival analysis, neither pollen taxon ($\chi^2_{18} = 5.59$, $p = 0.997$) nor bee size ($\chi^2_1 = 5.59$, $p = 0.44$) affected survival.

When we pooled taxa by species (*H. annuus*, *H. petiolaris*, *H. argophyllus*, *Solidago* spp.), species did not differ in their effects on *C. bombi* counts in a *post hoc* Tukey's HSD test. However, *C. bombi* cell counts in all pollen species were significantly lower than buckwheat and the wildflower pollen mix, by at least 60%.

In addition, we collected pollen both by hand and with honeybees for three taxa (*H. annuus* 'Black Oil Seed', 'Germany' and 'wild California'), allowing us to make direct within-species comparisons between collection methods. There were significant effects of collection method on *C. bombi* cell counts in two of the three direct comparisons but in opposite directions. Honeybee collection had higher *C. bombi* cell counts relative to hand collection in *H. annuus* 'Black Oil Seed' ($\chi^2_1 = 24.5$, $p < 0.001$, figure 2a) but

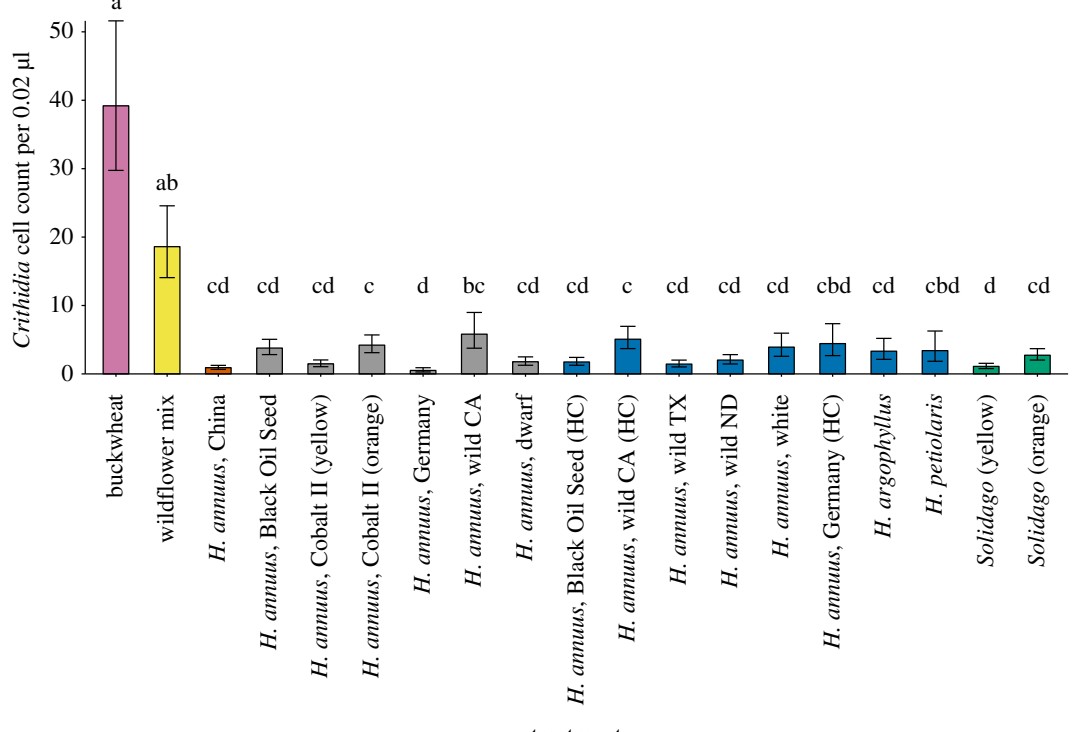

**Figure 1.** Mean raw *Crithidia* count per 0.02 µl ($\pm$ s.e.) for the 19 pollen taxa. Pollen treatments are: buckwheat (pink), wildflower mix (yellow), our positive control of *H. annuus* 'China' (orange), honeybee-collected taxa (grey), hand-collected taxa (blue) and *Solidago* spp. (green), which were honeybee-collected. Different letters associated with bars indicate statistically significant differences between pollen treatments after a *post hoc* Tukey's test. Full explanations for all taxa names are provided in electronic supplementary material, table S1; 'HC' refers to hand-collected for the three taxa where we had both honeybee and hand collection. Standard errors were calculated by back-transforming least-square means plus or minus least-square mean standard errors.

lower *C. bombi* counts in *H. annuus* 'Germany' ($\chi^2_1 = 6.26$, $p = 0.012$, figure 2*b*), and collection method had no effect in *H. annuus*, 'wild California' ($\chi^2_1 = 0.40$, $p = 0.5246$, figure 2*c*). When the 17 Asteraceae pollen taxa were grouped by collection method in an overall comparison, we found no statistically significant difference between collection methods on *C. bombi* counts ($\chi^2_1 = 0.95$, $p = 0.33$, figure 2*d*).

## 4. Discussion

Pollen from a wide variety of sunflowers reduced counts of the bumblebee gut pathogen *C. bombi* when compared to buckwheat pollen and wildflower mixed pollen. Bees fed *Solidago* spp. and *Helianthus* spp. pollens had 80–90% lower *C. bombi* cells compared to those that consumed buckwheat pollen. These results provide a much wider range of options for using sunflower pollen as a food supplement for managed bumblebees. Giacomini *et al.* [15] found that the intensity of *C. bombi* infection was lower in wild-caught workers when agricultural lands had more sunflower acreage. This study indicates that multiple sunflower cultivars or wild species could be used for pollen supplements or grown in pollinator-friendly plantings to help manage bee disease.

Although a wide range of sunflower pollen taxa dramatically reduced *C. bombi* infection in our study, sunflower pollen has low protein concentrations compared to other types of pollen [56]. Pollen with low protein can have multiple negative effects on bees, such as reducing hypopharyngeal gland size in honeybees [57], larval weight in *Bombus terrestris* [58], sweat bee offspring weight [59] and immune function in honeybees [60,61]. Although we found no differences in individual bee survival when fed sunflower, buckwheat or wildflower mixed pollen, we recommend that future work should compare the benefits and costs of sunflower pollen on bee performance, including reproduction, and ascertain the proportion of sunflower pollen in the diet that maximizes medicinal benefits while minimizing nutritional stress.

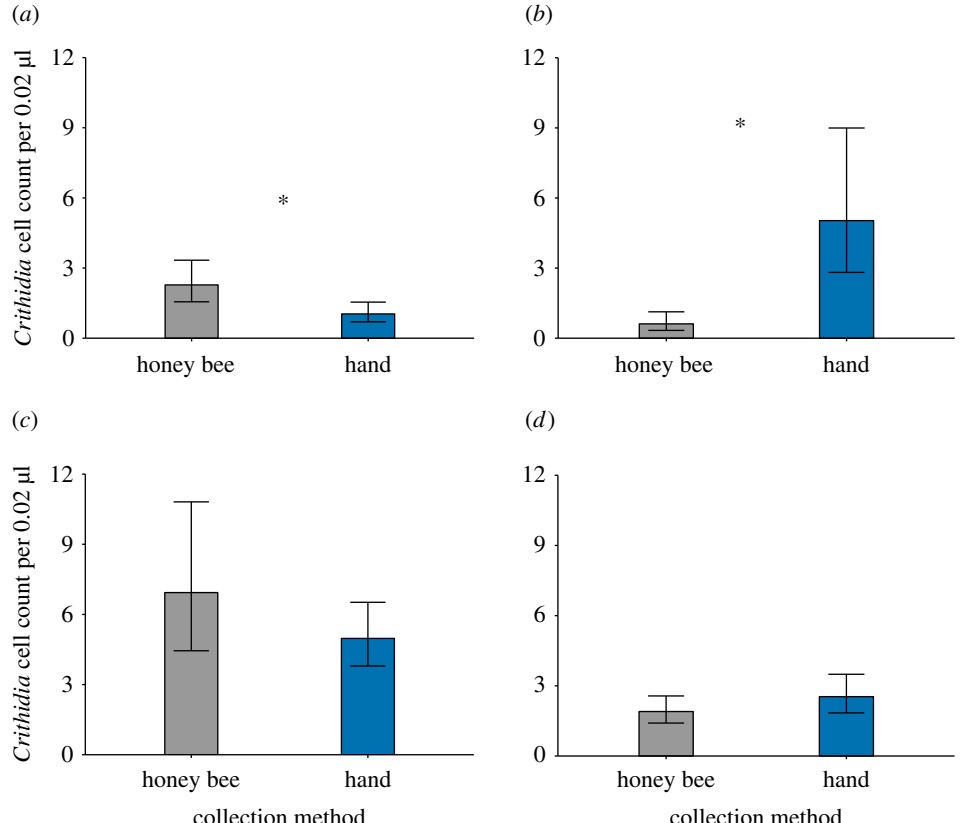

**Figure 2.** Comparison of hand- versus honeybee-collected pollen for (*a*) *H. annuus*, 'Black Oil Seed', (*b*) *H. annuus*, Germany, (*c*) *H. annuus*, wild California and (*d*) comparison pooled across all Asteraceae taxa used in the experiment (17 treatments). Asterisks (*) denote statistically significant differences between collection method. Standard errors were calculated by back-transforming least-square means plus or minus least-square mean standard errors.

In the temperate regions of North America, *Helianthus* spp. and *Solidago* spp. are common native plants [36,62]. Because *Solidago* is in a different tribe than *Helianthus* [63] but was equally effective at reducing *C. bombi*, it is possible that medicinal pollen is broadly widespread in the Asteraceae. Because Asteraceae are common components of many habitats and often bloom in mid- to late summer in temperate North America, this result could have important implications. By reducing parasite infections, these plant species could reduce one of the stressors affecting bumblebee populations. In *Bombus terrestris*, high *C. bombi* infection is negatively correlated with daughter queen emergence in wild colonies [30], and high infections can reduce early colony development by 40% when queens emerge from hibernation in spring [10]. Because *Solidago* spp. and many *Helianthus* spp. bloom in late summer and autumn, infected daughter queens could have an advantage if they forage on these floral resources before entering winter hibernation.

Previous studies assessing medicinal effects of sunflower floral rewards could not determine whether medicinal effects were due to pollen or nectar because they used honeybee-collected sunflower pollen or sunflower honey [15,16], both of which contain nectar and pollen. We compared the medicinal effect of hand- versus honeybee-collected pollen to ascertain whether the likely mechanism is due to a component of pollen or nectar. Surprisingly, in comparisons of hand- and honeybee-collected pollen within taxa, we found opposite results for different taxa. Within our three comparisons, we found all possible results: honeybee-collected pollen resulted in more *C. bombi* (figure 2*a*), less *C. bombi* (figure 2*b*) or no difference (figure 2*c*) compared to hand-collected pollen. In a larger comparison including all taxa, most of which were collected with only one of the two methods, there was no significant difference (figure 2*d*). Because we did not consistently find that honeybee-collected pollen (which contains nectar) reduced *C. bombi* counts relative to hand-collected pollen (which does not contain nectar), overall our results suggest that the main mechanism of reduced infection is due to some component of pollen rather than nectar.

Although most of our taxa had yellowish-orange pollen typical of many species in the *Helianthus* and *Solidago* [64], some of our taxa produced pollen in distinct colours of yellow (*Solidago* spp., and *H. annuus*

'Cobalt II'), orange (*Solidago* spp., *H. annuus* 'Cobalt II', and *H. annuus* 'China') or white (*H. annuus* 'white'). *Solidago* spp. and *H. annuus* 'Cobalt II' produced both yellow- and orange-coloured pollen, which were separated into two treatments (electronic supplementary material, table S1). We hypothesized that pigments might play a role in *C. bombi* suppression, since pigments are known to be biologically active and affect herbivores and bacteria [65,66]. For example, in *Petunia* hybrid flowers with white and blue petal sections, the white part of the petal was consumed more than the blue part by two generalist caterpillars, and larvae gained more weight feeding on white than blue tissue [67]. We found no support for the hypothesis that pollen colour affects *C. bombi* counts, suggesting pigments did not play a significant role in suppression. Yellow and orange pollen did not differ within a taxon, and *H. annuus* with white pollen did not differ from taxa with yellow or orange pollen in reducing *C. bombi* (figure 1).

Furthermore, while our results clearly demonstrate a substantive effect, the mechanism by which sunflower pollen reduces parasitism is unknown. Future research should address whether the medicinal quality of sunflower pollen is due to secondary chemistry, nutritional components or another mechanism, such as physical attachment of pollen to the parasite or the gut wall, preventing *C. bombi* from adhering to the gut wall [68]. Previous studies have shown that nectar secondary chemistry suppresses *C. bombi* [46,69] and honeybee immunity can be stimulated by the ingestion of some honeys [70]. Pollen proteins could also play a role. For example, the ragweed (*Ambrosia artemisiifolia*) pollen coat proteins trigger histamine production in humans as a defence response [71]. Finally, Asteraceae pollen is notable for its spines on the outer coat [72]. Given that *Crithidia* is a gut parasite that attaches to the hindgut wall [68], sunflower pollen could reduce parasitism by scouring the hindgut of parasite cells. Future work is needed to determine whether *H. annuus* and *Solidago* spp. pollen contain immune stimulants that induce upregulation of genes that reduce infection.

In conclusion, we found that sunflower and goldenrod pollen dramatically reduced the parasite *Crithidia bombi* in *Bombus impatiens*, compared to both a single-species pollen control and wildflower pollen mix. This study suggests that in addition to using sunflower and goldenrod to manage bee health in agroecosystems, these native North American species could be incorporated into natural ecosystems to manage *C. bombi* infection. Future work should address how widespread this medicinal effect is across the Asteraceae and the breadth of this medicinal effect for additional bee species and pathogens to make responsible recommendations for management practices.

Data accessibility. The datasets and R scripts supporting this article have been uploaded as part of the electronic supplementary material.

Authors' contributions. L.S.A. and R.E.I. conceived of the study with extensive design contributions from G.M.L. G.M.L. conducted the study and collected the data. G.M.L. analysed the data with assistance from L.A. G.M.L. wrote the manuscript with assistance from L.S.A. and all authors provided feedback on manuscript drafts.

Competing interests. We declare we have no competing interests.

Funding. The Lotta Crabtree foundation, Big Y Grocers, The Community Foundation of Western Massachusetts, USDA-AFRI 2013-02536 and USDA/CSREES MAS000411 and MAS00497 (Hatch) provided financial support. Any opinions, findings and conclusions or recommendations expressed in this material are those of the authors and do not necessarily reflect the views of the funding agencies.

Acknowledgements. We thank E. Palmer-Young and C. Sutherland for statistical advice, BioBest laboratories for donating bumblebee colonies, E. Dobbs and K. Bell for coordinating and conducting the DNA analysis for pollen taxa, A. Roy, K. Stinson, A. Averill, K. M. Connolly and L. Figueroa for comments and edits on the manuscript, K. Skyrme, J. Parrott and O. Ben-Shir for assisting with honeybee colonies, T. Rothchild for sewing tent covers, L. Rieseberg for seed germination protocols, K. Michaud, D. Delany, C. Grincavitch, E. Mann, C. Sergi, A. Zhao, L. Metz, T. Shaya, P. Deneen, J. Day, B. Joyce and A. Turkle for field and laboratory assistance, and Messa Farm, Laurenitis Farm and the Rattlesnake Gutter Trust of the use of their land to obtain pollen.

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
