## [Reviewer comments · Royal Society Open Science]

Review History

RSOS-190279.R0 (Original submission)

Review form: Reviewer 1

Is the manuscript scientifically sound in its present form?

Yes

Are the interpretations and conclusions justified by the results?

Yes

Is the language acceptable?

Yes

Is it clear how to access all supporting data?

Yes

Do you have any ethical concerns with this paper?

No

Have you any concerns about statistical analyses in this paper?

No

Recommendation?

Accept as is

Comments to the Author(s)

I was a reviewer on the previous version of this manuscript to Proceedings B and see that all of my concerns have been completely addressed. It is a fine contribution.

Review form: Reviewer 2

Is the manuscript scientifically sound in its present form?

Yes

Are the interpretations and conclusions justified by the results?

Yes

Is the language acceptable?

Yes

Is it clear how to access all supporting data?

Yes

Do you have any ethical concerns with this paper?

No

Have you any concerns about statistical analyses in this paper?

No

Recommendation?

Accept as is

Comments to the Author(s)

Dear Authors. Thanks for incorporating my previous, minor comments. I recommend publication with no further changes. Best regards.

Decision letter (RSOS-190279.R0)

12-Mar-2019

Dear Dr LoCascio:

It is a pleasure to accept your manuscript entitled "Pollen from multiple sunflower cultivars and species reduces a common bumble bee gut pathogen" in its current form for publication in Royal

Society Open Science. The comments of the reviewer(s) who reviewed your manuscript are included at the foot of this letter.

Please ensure that you send via email a zip folder containing:

- 1) an editable file version of your manuscript document (Word or Latex are preferred);
- 2) individual files for each figure;
- 3) individual files for each table;
- 4) a file containing captions for each figure and table.

on behalf of Prof Kevin Padian (Subject Editor).

Associate Editor Mr Andrew Dunn Comments to Author:

Associate Editor:

We're pleased to report that the reviewers who assessed your manuscript before it was transferred to Royal Society Open Science have indicated the paper is now ready for acceptance: congratulations! Thank you for the support of the journal, and we look forward to seeing your manuscript published in the near future.

Reviewer(s)' Comments to Author:

Reviewer: 1

Comments to the Author(s)

I was a reviewer on the previous version of this manuscript to Proceedings B and see that all of my concerns have been completely addressed. It is a fine contribution.

Reviewer: 2

Comments to the Author(s)

Dear Authors. Thanks for incorporating my previous, minor comments. I recommend publication with no further changes. Best regards.
